# Salinity tolerance, hyposaline stress recovery, and survival of the nemertean worm, *Carcinonemertes carcinophila* (Nemertea) in relation to its host, the Atlantic blue crab, *Callinectes sapidus*

**Alexandria K. Pomroy**, **Alexandra K. Schneider**, **Jeffrey D. Shields** *

Virginia Institute of Marine Science, The College of William & Mary, Gloucester Point, Virginia, United States of America

☉ All authors contributed equally to this work.
* jeff@vims.edu

## Abstract

*Carcinonemertes carcinophila* is a nemertean worm from a family of marine symbionts specialized in eating the eggs of decapod crustaceans. This species infests the Atlantic blue crab, *Callinectes sapidus*, a native to the Western Atlantic and Gulf of Mexico waters. Its host, the mature female blue crab, is euryhaline, migrating from low to high salinity waters during its adult life, rather than being exclusively marine. Unlike *C. carcinophila*, most species of marine nemerteans are stenohaline, living exclusively in high salinity waters. The salinity tolerance of *C. carcinophila* has not been well examined. This study used field-collected frequency data to assess the infestation intensity of nemerteans in relation to salinity regimes, and microcosm experiments to investigate the salinity tolerance and survival of *C. carcinophila* under hyposaline stress. These investigations also provide information on the nemertean's life history in relation to the spawning migration of female blue crabs. A multi-stage General Linear Model was used to test our hypothesized positive relationship between salinity and the probability of nemertean abundance on mature female crabs. Experiments confirmed that salinities of 20–30 psu were ideal for the survival of *C. carcinophila* and revealed the distinct ability of this species to acclimate rapidly to mesohaline conditions as low as 10 psu. This species was also able to withstand oligohaline stress (5 psu) for up to 39 hours. The wide range in salinity tolerance (10–30 psu) indicates that *C. carcinophila* has evolved to survive in similar euryhaline environments as its host. In addition to the wide salinity tolerance of the worm, the ability to withstand hyposaline stress indicates that rapid salinity changes in the blue crab's natural environment does not limit the reliability of *C. carcinophila* as a biomarker for the spawning history of blue crabs.

**Data availability statement:** Data are now available in figshare https://figshare. com/s/641d53ba42e7f369b6ae.

**Funding:** AKP received an NSF REU Fellowship (NSF #1950242 to R.D. Seitz) for summer internship support. This funding was not specifically for this project, but for student support in educational training. AKS received the Willard A. Van Engel Fellowship in support of her graduate training at VIMS, including her work on this project. AKS also received a Virginia Sea Grant Fellowship (#22203) in support of her graduate studies on the reproductive ecology of blue crabs, which included her work and supplies used for this project. The funders had no role in study design, data collection and analysis, decision to publish, or preparation of the manuscript.

**Competing interests:** The authors have declared that no competing interests exist.

## Introduction

The genus *Carcinonemertes* is comprised of symbiotic egg-predators, commonly found on decapod crustaceans. *Carcinonemertes carcinophila* (Kölliker) is a symbiont specific to the Atlantic blue crab, *Callinectes sapidus.* This decapod host lives in estuarine waters of the Western Atlantic and Gulf of Mexico [1–3]. The worms recruit and settle in the clutches of ovigerous crabs, where they feed on eggs and grow to maturity. During their maturation they grow from small, inconspicuous, white-colored worms to larger, bright pink, red, or orange-colored worms, after feeding on the yolk of embryos. They often congregate at the base of the host's abdomen because they are negatively phototactic [1]. Female and male nemerteans mate within the broods of their host and are commonly found entwined on setal strands of the host's pleopods [1,4]. During late-stage embryogenesis, the worms can migrate to the gills of the female crab, where they coil and encyst within a thin mucous sheath until subsequent brood production, when they may emerge and migrate back to the egg clutch [1].

Species of *Carcinonemertes* are typically stenohaline and exclusively infest marine decapods. However, the estuarine environment and migration patterns of blue crabs exposes *C. carcinophila* to a broader range of salinities, distinguishing them from other species of *Carcinonemertes* [1,5–7]. Chesapeake Bay in particular experiences extreme salinity fluctuations (0–28 psu) spatially and temporally due to variations in wet and dry seasons, and runoff from tributaries, such as the James and York rivers [8,9]. Blue crabs are well adapted to these fluctuations, migrating between high and low salinity regimes for growth and maturation [10,11]. Nonetheless, they require high salinities for viable embryogenesis [12]. Therefore, mature female crabs will migrate long distances, up to 250 km, from low to high salinity waters after mating; however, they do not migrate back into lower salinities and will remain in high salinity areas for oviposition [13–15]. Thus, blue crab migration may subject *C. carcinophila* to a broad range of salinities during their lifetime.

Female blue crabs produce one to three broods per spawning season in Chesapeake Bay, providing multiple opportunities for recruitment of larval nemerteans into crab broods [15]. This suggests the potential for accumulation of this nemertean over the lifetime of mature female crabs. Additionally, female crabs have a terminal (pubertal) molt; thus, once infected with *C. carcinophila*, female crabs likely retain all worms that recruit to them during their mature life, unless the worm is subject to lethal environmental conditions. This differs from host species with indeterminate growth that regularly molt and may naturally shed infested carapaces, potentially reducing nemertean infestations, at least in part [7,16].

The life history of both the female blue crab and *C. carcinophila* supports the use of these worms as biomarkers to define the spawning history of their female hosts [1,3,17,18]. Briefly, mature female crabs with large, pink, or orange-colored worms that are visible macroscopically in the gills, or both gills and brood, are considered multiparous. Egg bearing female crabs with large, pink or red colored worms in their brood, and not in their gills, are primiparous, or first-time spawners [19]. Mature female crabs with white, or light colored, immature worms in their gills have likely never spawned. This is because no embryos have been produced by the host for

the worms to feed on to transition to maturity, i.e., to achieve their bright pink or orange color. To support this method for determining female spawning history, we need to better understand the basic biology of *C. carcinophila,* particularly how ambient water conditions may limit the nemertean's survival and subsequent applicability as a biomarker to study host reproductive ecology.

The objectives of this study were to (1) experimentally determine the salinity tolerance of *Carcinonemertes carcinophila*, (2) investigate the worm's ability to recover from exposure to hyposaline environments, and (3) determine if salinity is a useful predictor for the presence and abundance of infestation of these worms on blue crabs *in situ*. We also hypothesized that increases in salinity would be positively associated with increases in nemertean presence, infestation, and abundance, because these worms are expected to thrive in euryhaline waters [19].

## Methods

### Host collection

Host blue crabs were collected from Cobb Bay, Eastern Shore, Virginia, using a 1-m scrape (toothless dredge) with a small mesh size (6.5 mm) bag. Worms from these crabs were used in the salinity tolerance portion of the study. Additional blue crab females and water quality data were also collected from May to September 2022, by the Virginia Institute of Marine Science (VIMS) Trawl Survey (hereafter called the VIMS trawl survey). The VIMS trawl survey is a fisheries independent survey that samples in the Virginia portion of Chesapeake Bay, and its tributaries [20]. Crabs collected by the VIMS trawl survey were collected from the mainstem of the Bay and their worms were used in the hyposaline recovery portion of this study.

### Salinity tolerance experiment

The salinity tolerance of *C. carcinophila* was investigated in a 5-day microcosm experiment. Worms were extracted from the gills of one female blue crab, collected from Cobb Bay, Eastern Shore, Virginia at approximately 30 psu. The dorsal portion of the crab's carapace was removed, and the gills excised from the body from their base and placed in finger bowls containing 25 psu filtered seawater. The worms' negative phototaxis was used to aid in extraction by using an overhead light source and gentle prodding of the gills with a blunt probe [18]. Worms were gently pipetted from the finger bowl and placed into a single well of a 6-well culture plate (Fisher) containing approximately 3 mL of 25 psu seawater for acclimation. The wells in the 6-well plates were approximately 35 mm in diameter, with a volume of 10 mL.

Worms were exposed to one of four salinity treatments: 5, 10, 20, 30 psu. Different salinities were made by mixing deionized water with salt water collected from the Bay with samples measured and adjusted with a refractometer (Fisher Scientific, optical refractometer for salinities 0–100 psu with automatic temperature correction). Six replicates were allocated per treatment (n = 24), and each replicate consisted of one worm held in one chamber of a four-compartment petri dish (9 cm diameter). Each of the four chambers were filled with one of the experimental salinities to investigate if nemerteans would actively seek less stressful salinities. In addition, a zero psu salinity control consisted of six worms individually housed in a 6-well plate. The wells were filled with deionized water, which acted as a proxy for zero salinity freshwater. Individuals were acclimated from 25 psu to treatment conditions by increasing or decreasing the salinity by 2 psu per hour. The start of acclimation was staggered for each treatment group to ensure that the experiment began at the same time for all treatments.

Once acclimated to the proper salinity, each worm was transferred to its own four-compartment petri dish; specifically, into the sector of the dish that held salt water with the same salinity the worm was acclimated to. All petri dishes were kept in the dark, except during the below assessments. Experimental units were kept at ambient temperature, approximately 22°C. Partial water changes (approximately 20%) were done daily.

Three qualitative metrics and one quantitative metric were used to assess nemertean responses to salinity over the 5-day trial. The qualitative metrics included worm body position, worm location in the petri dish, and response to a probe (referred to

as "probe response" hereafter). The quantitative metric was a negative phototactic response. Worm body position was categorized as lacking rigor, coiled, or elongated/folded. Worm location was categorized as in the center, at the floor-wall junction, or on the wall of the petri dish. Probe response was assessed using a trimmed, fine-haired paintbrush, to not injure the worms. Each nemertean was given three sequential prods with the paintbrush and the peristaltic response was qualified as either strong, mild, weak, or no response. Classification of the response also accounted for reaction time and amount of recoil in response to probing. Location, body position, and strength of response were subjective but interpreted by the same person (A.K. Pomroy) during each assessment. Assessments of these metrics were made every 24 hours.

Phototaxis was assessed qualitatively. A piece of copy paper, shaded in black ink, was placed under that half of the sector in which the nemertean was located. The paper was arranged so the nemertean was initially above the non-shaded side of the compartment. At the time of paper placement, a timer for 10 minutes was set and the time it took for the worm's eye spots to cross into the shaded region was recorded. Worms either did not respond to light stimulation, responded to light stimulation by moving in the direction of the shaded region and not crossing within the allotted 10 minutes, or responded to the light stimulation and crossed into the shaded region within the 10 minutes.

Worms were deemed in good condition when displaying an elongated or folded body position, when located at a floor-wall junction of a petri dish, when exhibiting a strong recoiling response to probing, or when exhibiting negative phototaxis. This is because nemerteans are thigmotactic, and responsive to mechanical stimuli. A positive response to tactile stimuli indicated they were alive and responsive [1]. Species within the genus typically exhibit negative phototaxis at maturity and thus congregate in dark spaces within the host's brood, such as at the base of embryo-bearing pleopods [1,4]. Thus, when an individual exhibited one or more of these behavioral responses, e.g., thigmotaxis or negative phototaxis, it was considered as exhibiting a positive response.

When a worm did not exhibit thigmotaxis or was unable to perform these behaviors, they were considered stressed/unresponsive. This occurred when a worm lacked rigor, was found in the center of the petri dish floor (i.e., not in contact with the walls), was unresponsive to probing, or did not exhibit negative phototaxis.

**Hyposaline recovery experiment**

The ability of *C. carcinophila* to recover from extreme hyposaline conditions (≤5 psu) after varying durations of exposure was investigated. At the end of the above exposure experiment, the nemerteans from the 20 psu and 30 psu treatments were acclimated back to 25 psu and transferred to wells in two 6-well plates (n = 11). They were kept at 25 psu, and room temperature conditions for 2 weeks. To increase the sample size (n = 14), three additional worms were extracted from the gills of one host female crab collected by the VIMS Trawl survey, at approximately 15 psu. Worms were kept in 6-well plate wells and salinity was stepped down from 25 psu to 5 psu, over 10 hours, at 2 psu per hour, for the start of the recovery experiment.

Assessment of worm condition began after all 14 worms spent 15 hours at 5 psu. Worm condition was assessed as described above, with the exception of omitting phototaxis assessments for this experiment. An additional metric, mucus production, was included. Mucus production functioned as a binary qualitative metric, monitoring the presence or absence of mucous sheath. The presence of the mucous sheath was assessed under a dissecting scope. The presence of a mucous sheath was considered a positive indicator of worm health because worms often reside in mucous sheaths while on the gills and eggs of their hosts [1]. Moreover, they abandon and remake these sheaths during their migrations between the host's brood and gills [1].

Worms in the short exposure treatment (n = 4) were exposed to 5 psu seawater for 15 hours, those in the moderate exposure treatment (n = 5) were exposed for 39 hours, and those in the long exposure treatment (n = 5) were exposed for 63 hours. At the end of the exposure period, the salinity was stepped back up to 25 psu, at 2 psu per hour. The worms were assessed for four days. On the final day of observation, worm condition was assessed after all worms were brought back to 25 psu for at least 15 hours. Water changes at the appropriate salinities were done daily.

## Nemertean presence and abundance in situ

To determine the presence and abundance of worms on their hosts, ovigerous female crabs were collected by the VIMS trawl survey from the mid and lower stem of Chesapeake Bay. Water quality parameters, specifically bottom salinity were measured by the VIMS trawl survey per tow. Once collected, the females were indexed by tow, kept on ice and brought back to the lab for assessment within 24–72 hours of catch. Egg development of each female's clutch was determined to be one of five potential stages, using brood presence and color: (1) no eggs, (2) orange eggs indicating early development, (3) brown eggs indicating mid development, (4) black eggs indicating late development, (5) egg remnants from recent eclosion [18–20]. Females were also inspected for nemertean infestation in their broods, if applicable, and their gills. For brood inspection, individual pleopods were removed from the body of the crab. Crab eggs were stripped from the setae, examined, and all visible worms were removed and counted. To examine the gills, the dorsal portion of the crab's carapace was removed, the gills were excised from the branchial chamber and placed in tap water to facilitate worm removal. Forceps were used to tease apart gill filaments, and remove and count all visible, encysted nemerteans.

## Data analysis

**Salinity tolerance and hyposaline recovery.** Data analysis and graphics were done in R, the statistical computing program [21]. All metrics were assessed by salinity treatment and day of experimentation. Given that the metrics were binomial in response (positive or negative), the standard error of the proportions was calculated as:

$$(p \times (1-p))/n)^{0.5} \tag{1}$$

where $p$, is the proportion of nemerteans that exhibited a given response and $n$ is the total number of worms in the sample [22,23]. One nemertean, originally assigned to the 30 psu treatment, was removed from the data analysis because it lacked ocelli and was deemed injured.

**Modeling nemertean presence and abundance.** To understand the relationship between worm presence, abundance, and salinity, a multi-stage general linear model (GLM) was used, with a binomial probability function modeling presence (logit link) and a negative binomial function modeling abundance. The binomial stage of the model accounts for excess zeros which are a result of ovigerous females potentially not yet having contracted nemerteans. Abundance was defined as the total number of worms on the eggs and gills of female blue crabs. In addition to salinity, egg stage was considered in models, due to known relationships between egg stage and worm presence and intensity [18]. Due to sample size and because eggs from stage 5 crabs would have likely just hatched, stage 4 and 5 were combined in all models. An interaction between egg development stage and salinity was included because blue crab eggs can tolerate lower salinities at earlier developmental stages (A. Schneider personal observation).

Five multi-stage models were hypothesized and tested (Table 1): **g1** is a baseline, intercept only model, **g2** is a model for worm abundance where the probability of having contracted nemerteans is dependent on an intercept only and the count of nemerteans is informed by salinity, **g3** is a model of worm abundance where both the probability of having contracted nemerteans and the count of nemerteans is informed by salinity, **g4** is a model for worm abundance where the probability of having contracted nemerteans is informed by salinity and the count of nemerteans is informed by salinity, egg development stage, and an interaction between the two, and **g5** is a model of worm abundance where both probability of having contracted nemerteans and the count of nemerteans is informed by salinity, egg development stage, and an interaction between the two. These five models were compared using Akaike Information Criterion (AIC) to determine the best fit for the relationship between salinity and nemertean infestation abundance. The zero-inflated negative binomial (ZINB) model was run with the MASS package, in R [21,22].

**Table 1. Hypothesized zero-inflated negative binomial models predicting nemertean abundance. Multistage model of nemertean worm presence modeled using a binomial generalized linear model (GLM), and abundance, modeled using a negative binomial GLM. The terms $\gamma_0$ and $\beta_0$ represent the model intercepts for the binomial, and negative binomial models, respectively. For models with egg stage the intercept represents the baseline condition, egg stage 1. The model with the lowest Akaike Information Criterion (AIC) is in bold.**

| Model | Binomial | Negative Binomial | AIC |
|---|---|---|---|
| g1 | $\gamma_0$ | $\beta_0$ | 1718.599 |
| g2 | $\gamma_0$ | $\beta_0 +$ salinity | 1708.289 |
| g3 | $\gamma_0 +$ salinity | $\beta_0 +$ salinity | 1673.889 |
| g4 | $\gamma_0 +$ salinity | $\beta_0 +$ salinity + egg stage$_i$ + (egg stage$_i$ × salinity) | 1644.183 |
| g5 | $\boldsymbol{\gamma_0}$ **+ salinity+egg stage$_i$+(egg stage$_i$× salinity)** | $\boldsymbol{\beta_0}$**+ salinity+egg stage$_i$+(egg stage$_i$ × salinity)** | 1635.811 |

An 89% predictive interval was used for both models. An alpha threshold of 0.05 was used for determining significance.

## Ethics statement

This research did not require review by institutional review boards. VIMS staff are not required to have a scientific collecting permit in the tidal waters of Virginia.

## Results

### Salinity tolerance

*Carcinonemertes carcinophila* exhibited a distinct survival pattern in response to salinity. The zero psu treatment (control) and 5 psu treatments were lethal for all worms. The worms in the 10 psu treatment performed poorly during the first three days, but their assessed responses improved by days four and five. The worms in the 20 psu and 30 psu environments consistently exhibited strong, positive responses over the 5 days by all metrics (Fig 1).

For the worms at 10 psu, the body position metric fluctuated, but by day five 100% of the 10 psu treatment nemerteans exhibited a body position indicative of good health (Fig 1A). The proportion of worms in 10 psu treatments recorded at the floor-wall junction of the plate (the location indicative of positive thigmotaxis) increased over the 5-day experiment: 16.7 (±15.2 SE) % on day one to 83 (±15.2 SE) % on days four and five (Fig 1B). The proportion of worms at 10 psu that exhibited negative phototaxis increased over the 5-day period, with 100% of worms in a 10 psu treatment exhibiting negative phototaxis on days four and five (Fig 1D). Worms at 20 psu responded to probing 100% of the time over the 5 days, with the majority (53±9.1 SE %) of those responses being strong (Fig 1C). The nemerteans at 20 psu exhibited negative phototaxis 70% of the time over the 5-days. The nemerteans at 30 psu had a strong response to the probe 76 (±8.5 SE) % of the time, over the 5-day period, with 100% of responses being strong on the fifth day (Fig 1C). Negative phototaxis was seen 80 (± 8.0 SE) % of the time over the 5-day period.

### Hyposaline recovery

Generally, the longer the nemerteans were subjected to hyposaline stress (≤5 psu), the more poorly they performed in health assessments. All the nemerteans in the short exposure treatment (15 hours) exhibited a body position indicating poor health on the first day. However, the percentage of worms in body positions indicating good health increased over the subsequent days from 25 (±21.6 SE) % on day two to 75 (±21.6 SE) % on day four (Fig 2A). There was no response to probing from worms in the short exposure group on the first day. On the second day, after being acclimated back to 25 psu, 50 (±25.0 SE) % of the worms had a weak probe response. On the third and fourth days, 75 (±21.6 SE) % of short exposure worms exhibited a probe response, the majority of which were mild or strong.

After the first 15 hours in a hyposaline environment, 80 (±17.8 SE) % of the moderate exposure worms exhibited a poor body position, and after 39 hours 100% of these nemerteans exhibited a poor body position. On the third day, after the moderate exposure treatment group had been acclimated back to 25 psu for 15 hours, 40 (± 21.9 SE) % of the worms

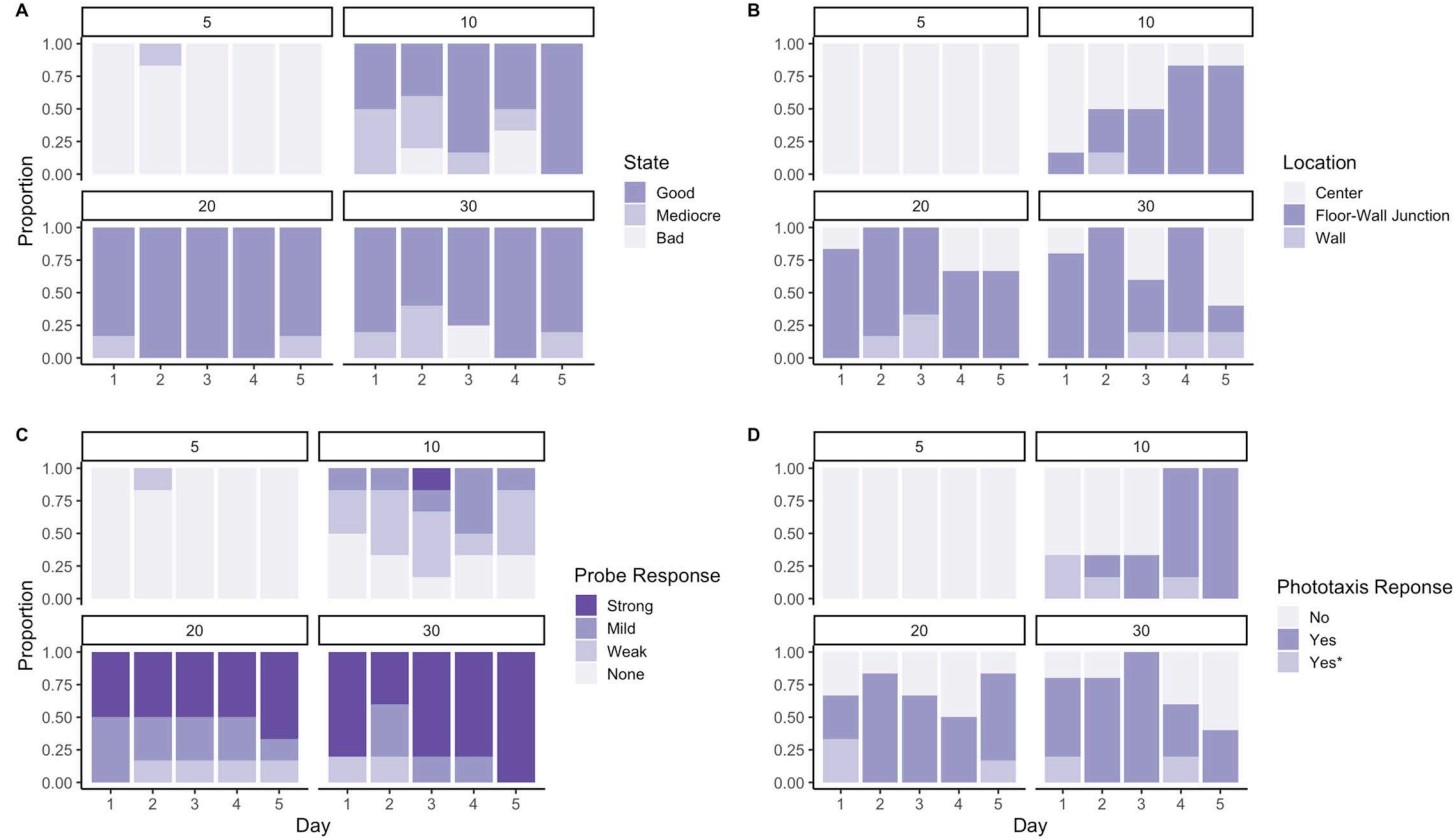

**Fig 1. Proportional responses of nemertean worms to salinity treatment over the 5-day microcosm experiment.** Headings indicate the salinity treatment in practical salinity units (psu), with 5 psu in top left, 10 psu top right, 20 psu bottom left, and 30 psu bottom right for all plots A through D. (A) State of body position of nemertean at time of observation. (B) Location of nemertean within well plate at time of observation. (C) Probe response, reported as a strong response (best indicator of health), mild response, weak response, or no response (poorest indicator of health). (D) Negative phototaxis response: "No" indicating no negatively phototactic movement; "Yes" indicating negatively phototactic movement into the shaded region within 10 minutes; "Yes*" indicating negatively phototactic movement towards the shaded region but did not move into the region within 10 minutes.

exhibited a good body position. On the last day, i.e., 24 hours later, 100% of the worms from the moderate exposure group were in a body position indicative of poor health (Fig 2A). All worms (100%) in the moderate exposure treatment had mucous sheaths present on the first two days. On day three, 80 (±17.8 SE) % had mucous sheaths present. By day four, 20 (±17.8 SE) % of the worm in the moderate exposure treatment had mucous sheaths (Fig 2B). Only two probe responses were seen from the moderate exposure treatment, both of which were weak, one on day three and one on day four (Fig 2C).

For the worms in the long exposure treatment, 80 (±17.8 SE) % were in a poor body position over the four days of experimentation (Fig 2A). On days two and three, 80 (±17.8 SE) % of had mucous sheaths still present. By day four, there was a 20% reduction in mucous sheath presence (Fig 2B). None of the worms from the long-duration exposure treatment showed a probe response during experimentation (Fig 2C).

## Modeling nemertean presence and abundance

The 198 female blue crabs assessed in this analysis were collected from 46 trawl survey stations. During the months of May-September 2022, salinity at trawl survey stations ranged from 15.6 psu to 30.1 psu, with 16.3 psu being the lowest

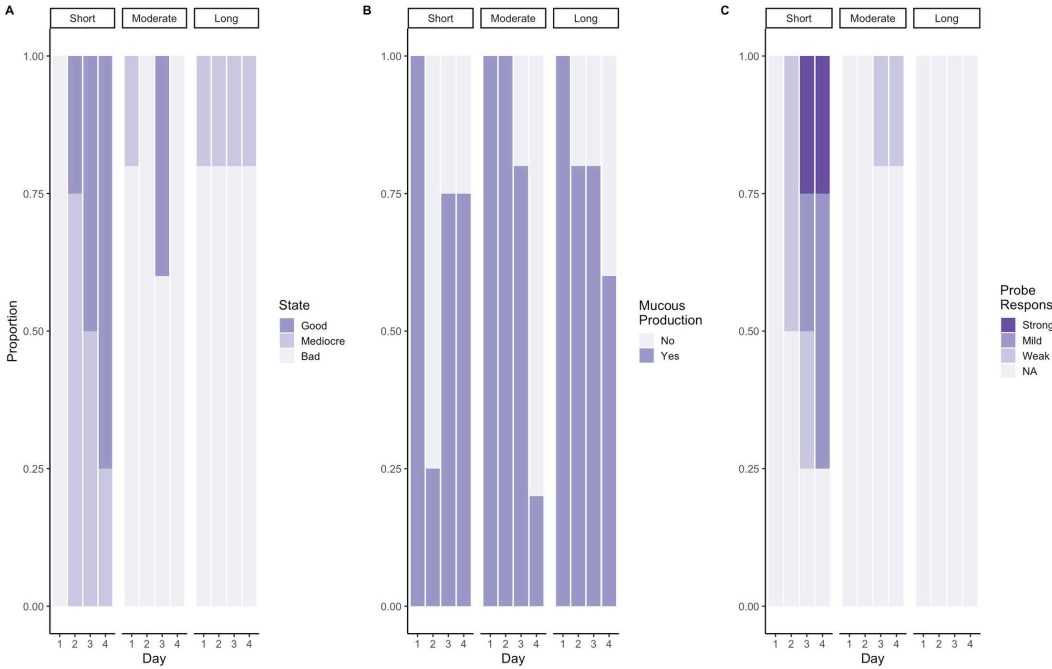

**Fig 2. Hyposaline stress experiment responses, reported as proportions, for each day.** Headings for each chart qualify the nemertean group's duration of hyposaline (5 psu) exposure. Short exposure worms were under hyposaline stress for 15 hours and acclimated back to a 25 psu after the day-1 check. Moderate exposure worms were under hyposaline stress for 39 hours and acclimated back to 25 psu after the day-2 check. The long exposure worms were under hyposaline stress for 63 hours and acclimated back to 25 psu after the day-3 check. (A) State of body position of nemertean at time of observation as defined in the methods. (B) Mucous sheath production as absence or presence. (C) Probe response reported as a strong response, mild response, weak response, or no response.

observed salinity where worms had infested crabs. The average salinity over the sampled time span was generally high, 24.2 psu (± 4.0 SD). Of the females assessed for worms, 61.6% of them had nemerteans on their brood or gills. The average abundance of nemerteans was 86.4 (± 9.7 SE) worms per female.

During the 2022 season (May-September), nemertean counts ranged from zero to >600 worms on individual crabs. The ZINB model accounting for salinity, egg stage, and the interaction, **g5**, had the lowest AIC score and selected for interpretation (Table 1). We see a 50% probability of nemertean infestation during stage 3 of development in 26.5 psu water. Salinity and the probability of infection by *C. carcinophila* displayed a strong positive association (Fig 3), however it varied by egg stage, becoming a weaker predictor of nemertean presence as crab eggs developed (Table 2). Similarly, there was generally a positive relationship between salinity and worm abundance, with the strength of that predictive relationship increasing over the course of embryo development (ZINB, Table 2).

## Discussion

In contrast to other species in the genus *Carcinonemertes*, *C. carcinophila* tolerates a wide range of salinities (10–30 psu), overlapping considerably with that of its host, *C. sapidus,* suggesting a tight coevolution of this species to the fluctuating habitats experienced by its host. This study supports 20–30 psu as the optimal salinity conditions for *C. carcinophila* because worms consistently exhibited several strong indicators of health when in the 20 and 30 psu treatments. However, the increase in positive responses over the 5-day period from nemerteans held in 10 psu seawater also demonstrates their ability to acclimate and survive on crabs at this lower salinity. More importantly, *C. carcinophila* can withstand and recover from short duration (up 39 hours) exposures to hyposaline stress as low as 5 psu, as might occur during a spring freshet.

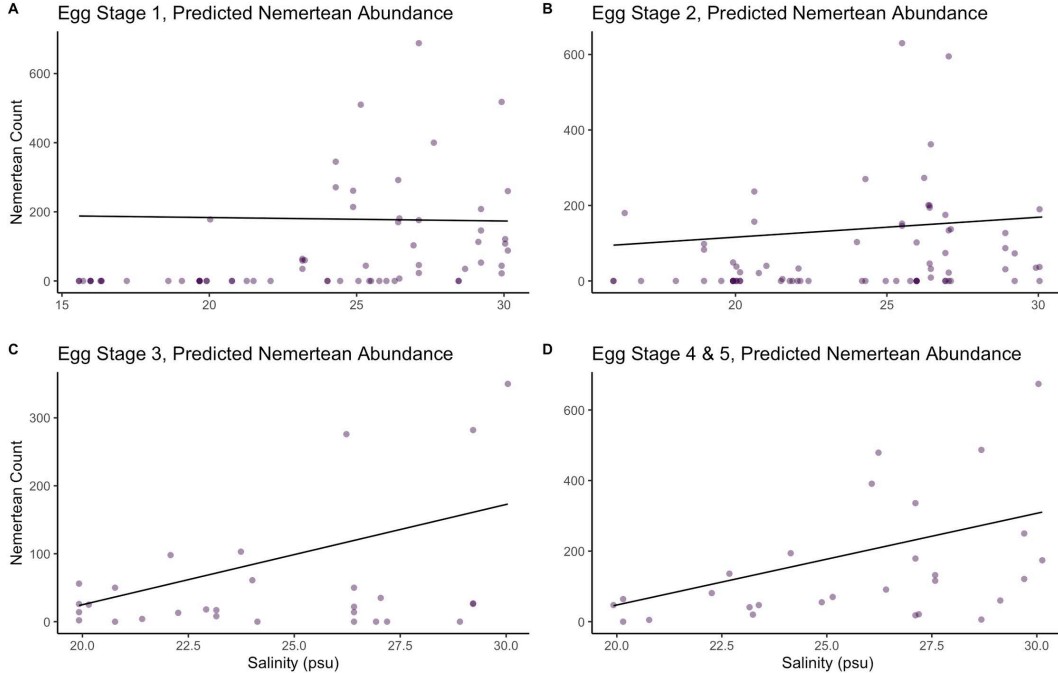

**Fig 3. The effect of salinity and egg stage on predicted nemertean abundance.** The line in the figure displays the negative binomial portion of the model, but all data is shown, including zeros. (A) ZINB model with salinity, egg stage 1, and their interaction as predictors. (B) ZINB model, with salinity, egg stage 2, and their interaction as predictors. (C) ZINB model, with salinity, egg stage 3, and their interaction as predictors. (D) ZINB model, with salinity egg stage 4 & 5, and their interaction as predictors (Table 1).

*Carcinonemertes carcinophila* salinity tolerance is in stark contrast to other species of *Carcinonemertes* that experience significant mortality at salinities ≤20 psu [24]. Furthermore, the worm's capacity to acclimate in mesohaline conditions and survive temporary oligohaline stress indicates that *C. carcinophila* is well adapted to survive and colonize blue crabs transiting mesohaline conditions during their adult migrations into higher salinities. The specimens used in this study were acquired from the Eastern Shore, a polyhaline environment, with an average surface salinity of 28 psu. Thus, the nemerteans used in experimentation likely came from an environment with a benthic salinity >24.0 psu and potentially more susceptible to hyposaline stress than their bayside counterparts, who reside in and migrate through more meso- and oligohaline waters. Clearly, *C. carcinophila* has a large range in salinity tolerance and ability to withstand and recover from hyposaline stress when present on their blue crab hosts.

Additionally, because C. carcinophila has a salinity tolerance over much of the range in salinity tolerance of blue crabs, female hosts likely do not experience worm die offs when exposed to ephemeral low salinity events. That is, there is little to no low salinity refuge from these egg-predators, unless it occurs over a prolonged time (>63hrs). Additionally, female hosts are unlikely to acquire *C. carcinophila* prior to maturation and migration to their spawning grounds, which is quite different from other species of nemertean such as *C. errans* that infest immature or non-ovigerous crabs in stenohaline oceanic waters [5,24].

The general vitality of *C. carcinophila* in 20–30 psu seawater, as well as its ability to acclimate to low salinity conditions, supports the use of this worm as a reliable biomarker for identifying the reproductive parity of female blue crabs in Chesapeake Bay. The improbable movement of mature female blue crabs from high to low saline water, the inability of nemerteans to re-infect another female host, the wide range of viable salinities for nemertean survival, and the lack of low salinity refuge for female hosts, all indicate that nemerteans stay on their host for the duration of the host's mature lifetime [1,4,5].

**Table 2. ZINB parameter estimates for both stages of model. The parameter estimates of the multi-stage model for predicted abundance of nemerteans on a host as salinity increases. SE, standard error.**

| Parameter | Estimate±SE | z value | P value |
|---|---|---|---|
| **Binomial** | | | |
| Intercept | 11.72±2.85 | 4.11 | 4.03e-05 |
| Salinity (psu) | −0.48±0.12 | −4.20 | 2.66e-05 |
| Egg Stage 2 | −7.88±3.28 | −2.40 | 0.02 |
| Egg Stage 3 | −17.49±4.99 | −3.51 | 0.00 |
| Egg Stages 4 & 5 | 27.49±63.12 | 0.436 | 0.66 |
| Salinity * Egg Stage 2 | 0.32±0.13 | 2.38 | 0.02 |
| Salinity * Egg Stage 3 | 0.66±0.20 | 3.37 | 0.00 |
| Salinity * Egg Stage 4 & 5 | −1.52±3.13 | −0.49 | 0.63 |
| **Negative Binomial** | | | |
| Intercept | 5.32±1.82 | 2.93 | 0.00 |
| Salinity (psu) | −0.01±0.07 | −0.08 | 0.93 |
| Egg Stage 2 | −1.42±2.14 | −0.67 | 0.50 |
| Egg Stage 3 | −6.06±2.28 | −2.66 | 0.01 |
| Egg Stage 4 & 5 | −5.27±2.43 | −2.17 | 0.03 |
| Salinity *Egg Stage 2 | 0.05±0.08 | 0.58 | 0.56 |
| Salinity * Egg Stage 3 | 0.20±0.09 | 2.30 | 0.02 |
| Salinity * Egg Stage 4 & 5 | 0.19±0.09 | 2.13 | 0.03 |
| Log(theta) | 0.16±0.12 | 1.27 | 0.20 |

Accordingly, we have shown that variable salinity, within the natural range of mature female blue crabs in Chesapeake Bay, does not interfere with the reliability of the worm's use as a biomarker. Furthermore, nemerteans on mature females in environments that are consistently less than 10 psu should not be used as biomarkers, due to the rapid disintegration of worms after more than 39 hours.

Our analytical models show that salinity is positively associated with probability of infestation and abundance of nemerteans. The relationship between salinity and nemertean infestation helps explain why the proportion of female of blue crabs infested with nemerteans increases as they move into higher salinities in Chesapeake Bay, and why females in mesohaline waters tend to be as heavily infested with nemerteans as those in euryhaline conditions.

We also found that accounting for the interaction between salinity and developmental stage of the crab's eggs improved the predictive power of the model. This is because crab embryogenesis requires higher salinities, and crabs further along in their embryogenesis have likely been in meso- and oligohaline waters for much longer, with their clutches susceptible to infestation, i.e., they have experienced a longer period of exposure to potential settlement by *C. carcinophila*. However, salinity becomes a less powerful predictor of nemertean presence as egg development continues. This is supported by Schneider et al., 2023 in which 96% of broods in late-stage development were observed to be infested [18]. That is, predicting the probability of nemertean presence on clutches with late-stage eggs (Stage 4 and 5) is difficult because most of these crabs already have worms present and therefore salinity loses its predictive power.

Ultimately these models provide statistical support for prevalence and abundance dispersion patterns seen in Chesapeake Bay and align with the outcome of our mesocosm experiments (see Figs 4 and 5). However, the low deviance (24.7%) explained by salinity in our models should encourage further studies using increased sample sizes and addition of other factors that may contribute to nemertean distributions, such as host origin, nemertean colonization patterns, worm lifespan relation to host life history, and nemertean temperature thresholds. For example, the variability in dispersion

 

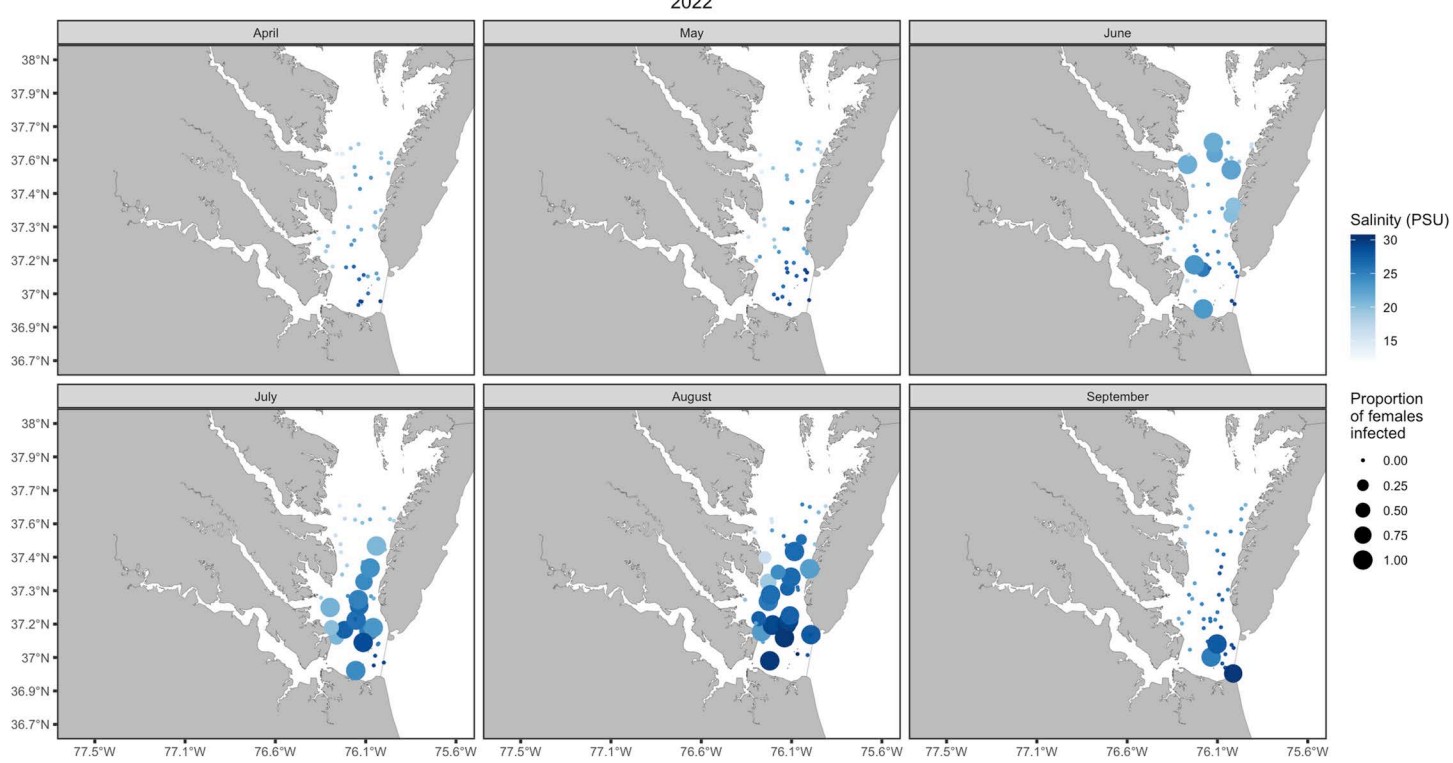

**Fig 4. Spatial distribution of nemertean presence on female blue crabs from Chesapeake Bay.** Locations of VIMS Trawl Survey tows, May 2022 through September 2022. The proportion of captured egg bearing females infected with *C. carcinophila* (circle size), and the salinity (psu) of the capture site at the time of tow (circle shade of blue).

patterns of prevalence and abundance could be the result of freshwater input, seasonal rain, or fishing pressure on the host [25]. This study identifies unique coevolutionary and life history patterns of *C. carcinophila*, independent of and in relation to, its symbiosis with *C. sapidus*; namely, its distinctly wide and compliant salinity tolerance and survival, in comparison to other species within this genus of symbionts.

## Acknowledgments

We thank Professor Randy Chambers and Dr. Jenny Rahn for their guidance and support. The crew and captains of the VIMS Trawl Survey, particularly Wendy Lowery, helped by obtaining crabs. Special thanks to Challen Hyman, Violet Johnston, Jainita Patel, Katie Knick, Mike Seebo, and the VIMS Community Ecology & Conservation Lab for their guidance and help with dissections. The statements, findings, conclusions, and recommendations are those of the authors and do not necessarily reflect the views of Virginia Sea Grant, NOAA.

## Author contributions

**Conceptualization:** Alexandria K. Pomroy, Alexandra K. Schneider, Jeffrey D. Shields.

**Formal analysis:** Alexandria K. Pomroy.

**Investigation:** Alexandria K. Pomroy, Alexandra K. Schneider.

**Methodology:** Alexandria K. Pomroy, Alexandra K. Schneider, Jeffrey D. Shields.

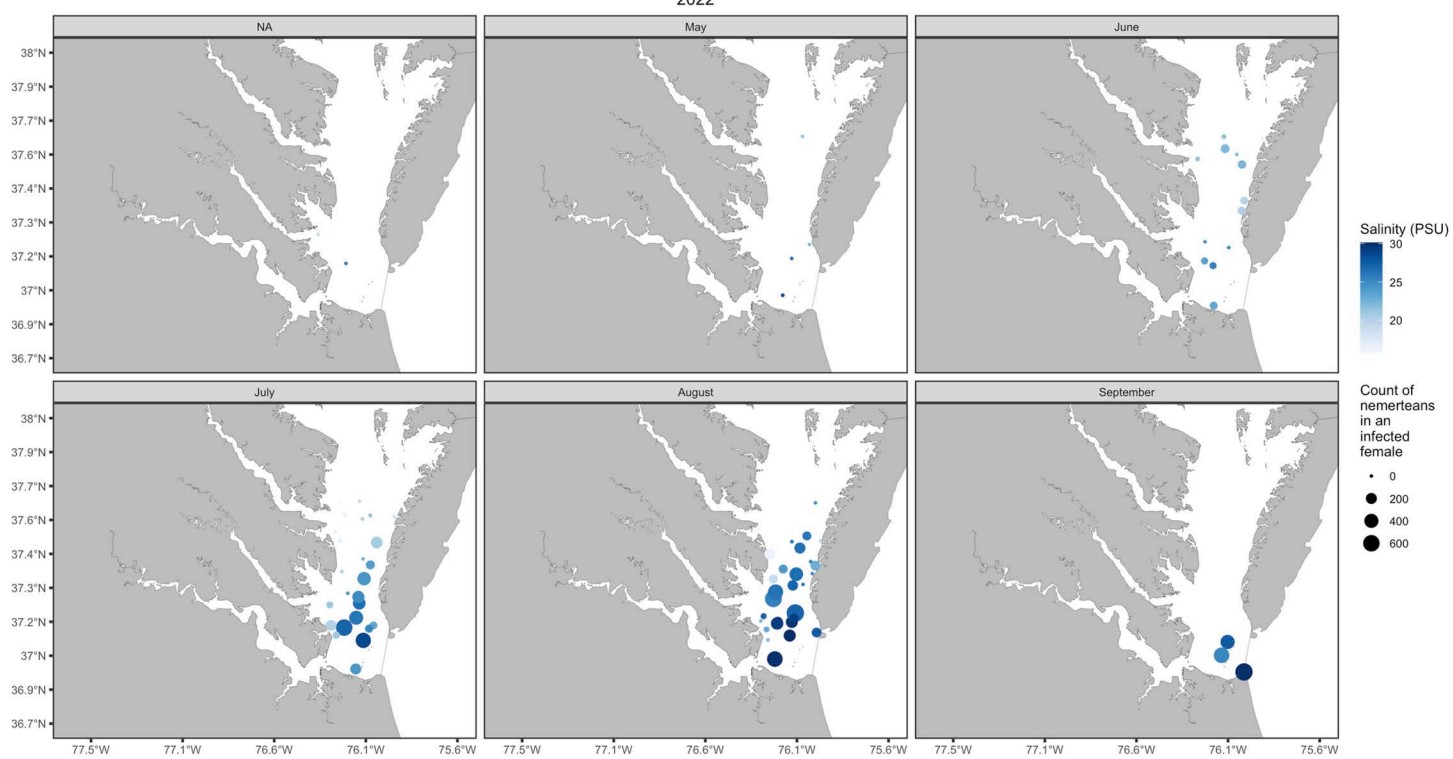

**Fig 5. Spatial distribution of nemertean abundance on female blue crabs from Chesapeake Bay.** Locations of Virginia Institute of Marine Science Juvenile Fish Trawl Survey tows, May 2022 through September 2022, and the maximum number of *C. carcinophila* in a caught, infested female (circle size), and the salinity (psu) at the site at the time of tow (circle shade of blue).

**Project administration:** Jeffrey D. Shields.

**Resources:** Alexandra K. Schneider, Jeffrey D. Shields.

**Supervision:** Alexandra K. Schneider.

**Validation:** Alexandra K. Schneider.

**Visualization:** Alexandria K. Pomroy.

**Writing – original draft:** Alexandria K. Pomroy.

**Writing – review & editing:** Alexandra K. Schneider, Jeffrey D. Shields.

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
