## [Decision Letter · Decision Letter 0]

*Carcinonemertes carcinophila* (Nemertea) in relation to its host, the Atlantic blue crab, *Callinectes sapidus*

Dear Dr. Shields,

**The reviewer 1 requests Major Revisions of your manuscript. You will find Reviewer 1 decisions requesting major revisions. I agree with him, but I think these revisions will be easy to take up and integrate into your manuscript.**

**Regarding your manuscript, I would suggest to add more references and recents on the effect of salinity on the blue crab Callinectes sapidus to contextualised better your study.**

We look forward to receiving your revised manuscript.

Kind regards,

Guillaume Marchessaux, Ph.D.

Academic Editor

PLOS ONE

**Journal Requirements:**

1. When submitting your revision, we need you to address these additional requirements. Please ensure that your manuscript meets PLOS ONE's style requirements, including those for file naming. The PLOS ONE style templates can be found at https://journals.plos.org/plosone/s/file?id=wjVg/PLOSOne_formatting_sample_main_body.pdf and https://journals.plos.org/plosone/s/file?id=ba62/PLOSOne_formatting_sample_title_authors_affiliations.pdf 2. Thank you for stating the following financial disclosure: AKS, Willard A. Van Engel Fellowship at VIMSAKS, Virginia Sea Grant #222030AKP, REU Fellowship, NSF #1950242Funding not specifically for this project, but for student support in educational training.   Please state what role the funders took in the study.  If the funders had no role, please state: "The funders had no role in study design, data collection and analysis, decision to publish, or preparation of the manuscript." If this statement is not correct you must amend it as needed. Please include this amended Role of Funder statement in your cover letter; we will change the online submission form on your behalf. 3. Thank you for stating the following in the Acknowledgments Section of your manuscript: We thank Professor Randy Chambers and Dr. Jenny Rahn for their guidance and support. The crew and captains of the VIMS Trawl Survey, particularly Wendy Lowery, helped by obtaining crabs. The VIMS trawl survey is funded by VIMS, The Virginia Marine Resources Commission, and U.S. Fish and Wildlife. Special thanks to Challen Hyman, Violet Johnston, Jainita Patel, Katie Knick, Mike Seebo, and the VIMS Community Ecology & Conservation Lab for their guidance and help with dissections. This work was supported in part by an NSF REU grant 1950242 to R.D. Seitz, Willard A. Van Engel Fellowship, and the Virginia Sea Grant Graduate Fellowship under Grant #V721500, Virginia Sea Grant College Program Project No. 222030, from the National Oceanic and Atmospheric Administration’s (NOAA) National Sea Grant College Program, U.S. Department of Commerce. The statements, findings, conclusions, and recommendations are those of the authors and do not necessarily reflect the views of Virginia Sea Grant, NOAA, or the U.S. Department of Commerce. We note that you have provided funding information that is not currently declared in your Funding Statement. However, funding information should not appear in the Acknowledgments section or other areas of your manuscript. We will only publish funding information present in the Funding Statement section of the online submission form. Please remove any funding-related text from the manuscript and let us know how you would like to update your Funding Statement. Currently, your Funding Statement reads as follows: AKS, Willard A. Van Engel Fellowship at VIMSAKS, Virginia Sea Grant #222030AKP, REU Fellowship, NSF #1950242Funding not specifically for this project, but for student support in educational training.  Please include your amended statements within your cover letter; we will change the online submission form on your behalf. 4. In the online submission form, you indicated that your data will be submitted to a repository upon acceptance.  We strongly recommend all authors deposit their data before acceptance, as the process can be lengthy and hold up publication timelines. Please note that, though access restrictions are acceptable now, your entire minimal dataset will need to be made freely accessible if your manuscript is accepted for publication. This policy applies to all data except where public deposition would breach compliance with the protocol approved by your research ethics board. If you are unable to adhere to our open data policy, please kindly revise your statement to explain your reasoning and we will seek the editor's input on an exemption. 5. When completing the data availability statement of the submission form, you indicated that you will make your data available on acceptance. We strongly recommend all authors decide on a data sharing plan before acceptance, as the process can be lengthy and hold up publication timelines. Please note that, though access restrictions are acceptable now, your entire data will need to be made freely accessible if your manuscript is accepted for publication. This policy applies to all data except where public deposition would breach compliance with the protocol approved by your research ethics board. If you are unable to adhere to our open data policy, please kindly revise your statement to explain your reasoning and we will seek the editor's input on an exemption. Please be assured that, once you have provided your new statement, the assessment of your exemption will not hold up the peer review process. 6. Please include your full ethics statement in the ‘Methods’ section of your manuscript file. In your statement, please include the full name of the IRB or ethics committee who approved or waived your study, as well as whether or not you obtained informed written or verbal consent. If consent was waived for your study, please include this information in your statement as well. 7. We note that Figures 4 and 5 in your submission contain map images which may be copyrighted. All PLOS content is published under the Creative Commons Attribution License (CC BY 4.0), which means that the manuscript, images, and Supporting Information files will be freely available online, and any third party is permitted to access, download, copy, distribute, and use these materials in any way, even commercially, with proper attribution. For these reasons, we cannot publish previously copyrighted maps or satellite images created using proprietary data, such as Google software (Google Maps, Street View, and Earth). For more information, see our copyright guidelines: http://journals.plos.org/plosone/s/licenses-and-copyright. We require you to either present written permission from the copyright holder to publish these figures specifically under the CC BY 4.0 license, or remove the figures from your submission: a. You may seek permission from the original copyright holder of Figures 4 and 5 to publish the content specifically under the CC BY 4.0 license.   We recommend that you contact the original copyright holder with the Content Permission Form (http://journals.plos.org/plosone/s/file?id=7c09/content-permission-form.pdf) and the following text:“I request permission for the open-access journal PLOS ONE to publish XXX under the Creative Commons Attribution License (CCAL) CC BY 4.0 (http://creativecommons.org/licenses/by/4.0/). Please be aware that this license allows unrestricted use and distribution, even commercially, by third parties. Please reply and provide explicit written permission to publish XXX under a CC BY license and complete the attached form.” Please upload the completed Content Permission Form or other proof of granted permissions as an "Other" file with your submission. In the figure caption of the copyrighted figure, please include the following text: “Reprinted from [ref] under a CC BY license, with permission from [name of publisher], original copyright [original copyright year].” b. If you are unable to obtain permission from the original copyright holder to publish these figures under the CC BY 4.0 license or if the copyright holder’s requirements are incompatible with the CC BY 4.0 license, please either i) remove the figure or ii) supply a replacement figure that complies with the CC BY 4.0 license. Please check copyright information on all replacement figures and update the figure caption with source information. If applicable, please specify in the figure caption text when a figure is similar but not identical to the original image and is therefore for illustrative purposes only.The following resources for replacing copyrighted map figures may be helpful: USGS National Map Viewer (public domain): http://viewer.nationalmap.gov/viewer/The Gateway to Astronaut Photography of Earth (public domain): http://eol.jsc.nasa.gov/sseop/clickmap/Maps at the CIA (public domain): https://www.cia.gov/library/publications/the-world-factbook/index.html and https://www.cia.gov/library/publications/cia-maps-publications/index.htmlNASA Earth Observatory (public domain): http://earthobservatory.nasa.gov/Landsat:
http://landsat.visibleearth.nasa.gov/USGS EROS (Earth Resources Observatory and Science (EROS) Center) (public domain): http://eros.usgs.gov/#Natural Earth (public domain): http://www.naturalearthdata.com/

**Additional Editor Comments:**

Dear authors,

Thank you for submitting your article to PLOS ONE. 

You will find Reviewer 1 decisions requesting major revisions. I agree with him, but I think these revisions will be easy to take up and integrate into your manuscript.

Regarding your manuscript, I would suggest to add more references and recents on the effect of salinity on the blue crab Callinectes sapidus to contextualised better your study.

Best regards,

Guillaume Marchessaux

Reviewers' comments:

Reviewer's Responses to Questions

**Comments to the Author**

1. Is the manuscript technically sound, and do the data support the conclusions?

Reviewer #1: Yes

2. Has the statistical analysis been performed appropriately and rigorously?

Reviewer #1: Yes

3. Have the authors made all data underlying the findings in their manuscript fully available?

Reviewer #1: Yes

4. Is the manuscript presented in an intelligible fashion and written in standard English?

Reviewer #1: Yes

**Reviewer #1:**  This study investigates the impact of salinity on nemertean parasites within blue crabs, a species of significant ecological and economic value. Key strengths include its focus on host-parasite interactions and how environmental factors like salinity influence them, particularly in euryhaline organisms. The research also contributes to understanding parasite tolerance to hyposaline stress, which is important for predicting their distribution. Given blue crabs' commercial importance, the study's findings have applied relevance for fisheries management. The controlled microcosm experiments and the inclusion of behavioral observations, such as phototaxis and probe response, enhance the study's depth. However, potential weaknesses exist. These include the need for better control over female crab origin to minimize host condition variability, the qualitative nature of behavioral metrics that could be improved with quantitative tracking, and potentially low sample sizes in certain treatments, which might limit statistical power.

Specific comments

the hypotheses overlap significantly and could be reworded to eliminate redundancy. Both essentially propose that salinity is positively correlated with nemertean presence, with the first focusing on infestation probability and the second on presence and abundance.

To improve clarity, these hypotheses could be combined into a single statement:

"We hypothesize that salinity will be positively correlated with the presence, abundance, and probability of nemertean infestation, as these organisms are expected to thrive in euryhaline waters."

Suggested Revisions for Specific Sections:

Host Collection: Please specify the original salinity/salinities.

Lines 108, 170, and 295: When referring to "hyposaline stress," clarify which specific salinity levels were used.

Line 112: Indicate the number of female crabs used in the experiment.

Line 117: Reword for formality: “containing approximately 3 mL.” same through the paper

Line 120: Explain how the experimental water was prepared, including details on the device used to measure salinity.

Line 162: The phrase “when an individual exhibited one or more of these taxes” is unclear. consider rewording to “behavioral responses”

Repeated use of "healthy": Consider simplifying or defining the criteria concisely.

Line 139: Clarify the categorization of worm location within the petri dish to distinguish positional differences more explicitly. there is potential for subjectivity in this response.

Line 175: When mentioning crabs from the lower Chesapeake Bay, include the recorded salinity for the experiments

Lines 183-184: Provide a reference to support the consideration of mucus production as a health indicator.

Line 195: Specify the total number of ovigerous females examined.

Line 224: Include female as a random factor in the experimental design and modeling to account for individual variation.

Controlling Female Crab Origin: To ensure statistical and experimental consistency, the origin of female crabs should be explicitly stated in the experimental design. This would help control for potential variability in host condition and exposure history.

**Do you want your identity to be public for this peer review?** For information about this choice, including consent withdrawal, please see our Privacy Policy

Reviewer #1: No

---

## [Author Response · Author response to Decision Letter 1]

29 May 2025

A response to reviewers comments file has been uploaded.

---

## [Editor Report · Decision Letter 1]

Salinity tolerance, hyposaline stress recovery, and survival of the nemertean worm, *Carcinonemertes carcinophila* (Nemertea) in relation to its host, the Atlantic blue crab, *Callinectes sapidus*

PONE-D-25-02790R1

Dear Dr. Jeffrey D. Shields,

Thank you for your interesting paper and for the revisions you made. We are pleased to inform you that your manuscript has been judged suitable for publication and will be formally accepted for publication once it meets all outstanding technical requirements.

Kind regards,

Guillaume Marchessaux, Ph.D.

Academic Editor

PLOS ONE
---

## [Editor Report · Acceptance letter]

PONE-D-25-02790R1

PLOS ONE

Dear Dr. Shields,

I'm pleased to inform you that your manuscript has been deemed suitable for publication in PLOS ONE. Congratulations! Your manuscript is now being handed over to our production team.

Kind regards,

on behalf of

Dr. Guillaume Marchessaux

Academic Editor

PLOS ONE